# Highly parallelized droplet cultivation and prioritization of antibiotic producers from natural microbial communities

Lisa Mahler[1,2], Sarah P Niehs[3], Karin Martin[1], Thomas Weber[1], Kirstin Scherlach[3], Christian Hertweck[2,3], Martin Roth[1], Miriam A Rosenbaum[1,2]*

[1]Bio Pilot Plant, Leibniz Institute for Natural Product Research and Infection Biology - Hans Knöll Institute, Jena, Germany; [2]Faculty of Biological Sciences, Friedrich Schiller University, Jena, Germany; [3]Biomolecular Chemistry, Leibniz Institute for Natural Product Research and Infection Biology - Hans Knöll Institute, Jena, Germany

**Abstract** Antibiotics from few culturable microorganisms have saved millions of lives since the 20th century. But with resistance formation, these compounds become increasingly ineffective, while the majority of microbial and with that chemical compound diversity remains inaccessible for cultivation and exploration. Culturing recalcitrant bacteria is a stochastic process. But conventional methods are limited to low throughput. By increasing (i) throughput and (ii) sensitivity by miniaturization, we innovate microbiological cultivation to comply with biological stochasticity. Here, we introduce a droplet-based microscale cultivation system, which is directly coupled to a high-throughput screening for antimicrobial activity prior to strain isolation. We demonstrate that highly parallelized in-droplet cultivation starting from single cells results in the cultivation of yet uncultured species and a significantly higher bacterial diversity than standard agar plate cultivation. Strains able to inhibit intact reporter strains were isolated from the system. A variety of antimicrobial compounds were detected for a selected potent antibiotic producer.

*For correspondence: miriam.rosenbaum@leibniz-hki.de

Competing interests: The authors declare that no competing interests exist.

## Introduction

The microbial diversity in environmental habitats provides a rich resource for medically relevant substances (**Moloney, 2016**), and fortunately, it is far from being exhaustively mapped (**Lloyd et al., 2018**). Only 1–15% of the bacterial diversity are cultivable under standard laboratory conditions (**Lok, 2015**), which promises further revolutionizing compounds to be discovered in the remaining unexplored majority. The question is: How can we unlock the hidden biosynthetic potential of so far uncultured microbes? Initial continuous propagation of cultures is mandatory to study the chemical and biological characteristics of their respective natural products. Thus, elucidation and application of appropriate growth conditions are required for cell replication of species that are recalcitrant to cultivation.

To narrow the gap between the naturally occurring complexity of microbial communities and the limited diversity of culture collections, several new techniques have been developed. For instance, the concentration and composition of media have been adapted for a large number of oligotrophic uncultured species (**Olsen and Bakken, 1987**; **Janssen et al., 2002**; **Sait et al., 2002**; **Joseph et al., 2003**; **Stevenson et al., 2004**; **da Rocha et al., 2010**). Other techniques mimic the conditions in situ in more detail by supplying extracts derived from the environment containing unspecified mixtures of macro- and micronutrients and other growth factors (**Olsen and Bakken, 1987**; **Rappé et al., 2002**; **Zengler et al., 2002**; **Sipkema et al., 2011**). The most advanced development in this direction is to provide direct contact with the environment (**Ben-Dov et al., 2009**; **Kaeberlein et al.,**

**eLife digest** Antibiotics are chemicals derived from microorganisms that can kill the bacteria that harm human health. In the 20[th] and 21[st] centuries antibiotics saved millions of lives, but new strains of dangerous bacteria that cannot be killed by antibiotics, known as antibiotic resistant strains, are becoming more frequent. Most antibiotics are produced by only a small group of microorganisms, but many more microorganisms exist in nature. So it is possible that microorganisms outside this small group can produce different antibiotics that are effective against antibiotic resistant strains.

Unfortunately, finding the microorganisms that produce these alternative antibiotics is challenging because researchers do not know which bacteria are producing these molecules and how to grow these microorganisms in the laboratory. To solve this problem, Mahler et al. developed a new method for growing a new subset of microorganisms in the laboratory. This would allow researchers to study the new microorganisms under controlled conditions, and determine whether any of the substances they produce have antibiotic properties.

Mahler et al. generated tiny droplets that could only contain a single cell of a microorganism, so each microbe could grow alone in its own protected environment. Using this approach, it was possible to grow completely different types of microorganisms than with traditional techniques, and keep them isolated from each other. This allowed each different species of microbe to be screened for antimicrobial activity, allowing the identification of chemicals that could potentially be developed into new antibiotics. This new method is automated and miniaturized, paving the way for growing many more cells in few hours, with very low material and space requirements.

These results showcase a way of growing new types of microorganisms in the laboratory, making it easier and faster to study them and determine what chemicals they produce. Understanding a greater variety of microorganisms in detail can help identify new chemicals for industrial applications, including new ways of combating infections.

*2002*; *Bollmann et al., 2007*; *Sizova et al., 2012*), which proved highly productive in culturing species regarded as unculturable.

A complementary trend is the miniaturization of culture vessels to save material and time while increasing throughput (*Zengler et al., 2002*; *Ben-Dov et al., 2009*; *Jiang et al., 2016a*; *Terekhov et al., 2017*; *Cao et al., 2017*; *Watterson et al., 2020*; *Villa et al., 2020*; *Martin et al., 2003*; *Ingham et al., 2007*; *Nichols et al., 2010*; *Gao et al., 2013*; *Jung et al., 2014*; *Ma et al., 2014a*; *Ma et al., 2014b*; *Tandogan et al., 2014*). Thereby, the chance of finding non-dormant, culturable variants of naturally occurring species, as stochastic events (*Epstein, 2013*), rises. These microscale culture techniques enable partitioning of complex bacterial communities at the single-cell level, which allows various species to grow at their own speed without competition for space or nutrients. At the same time, the miniaturized vessels that confine single cells are small and densely packed to establish a highly parallelized cultivation. The reported microscale approaches can be divided into methods that arrange the compartments into arrays (*Martin et al., 2003*; *Ingham et al., 2007*; *Nichols et al., 2010*; *Gao et al., 2013*; *Jung et al., 2014*; *Ma et al., 2014a*; *Ma et al., 2014b*; *Tandogan et al., 2014*; *Jiang et al., 2016a*; *Cao et al., 2017*) and techniques that keep the compartments not fixed to positions but rather mobile within the bulk population (*Zengler et al., 2002*; *Ben-Dov et al., 2009*; *Terekhov et al., 2017*; *Watterson et al., 2020*; *Villa et al., 2020*). The clear advantage of arrays is the possibility to address the same culture several times since the compartments are identified by their coordinates.

However, array technologies also come with inherent disadvantages. First, space is lost by introducing a distance between array positions and omitting the expansion in the third dimension. Arrays, like microtiter plates or miniaturized versions thereof, have so far only vessels arranged in a 2D plane. Second, fast automation is simpler with moving compartments and the continuous flow of compartments can be easily adjusted in speed and direction (*Vincent et al., 2010*). Consequently, without arrays, higher absolute numbers of compartments can be generated and processed per unit of time.

We propose an integrated cultivation and screening strategy based on surfactant-stabilized microfluidic droplets in a perfluorinated oil phase. Microfluidic droplets provide a genotype–phenotype linkage also for secreted products, meaning that those compounds stay confined with the cell from which they originated. Droplets can be stored in bulk, allowing to generate and incubate large droplet populations under controlled aerobic conditions. Furthermore, we implemented a high-throughput screening for antimicrobial products using bacterial reporter strains in droplets (*Mahler et al., 2018*). We verified our cultivation and screening strategy by culturing a complex bacterial community derived from a soil sample in a growth substrate designed accordingly and compared the cultivation outcome to classical agar plate cultivation for the same sample. For one of the isolates that showed antimicrobial activity in the droplet screening, we identified and characterized the secreted metabolites.

## Results

### Concept of cultivation in pL-droplets and workflow

We used a droplet-microfluidic platform to singularize, cultivate, and screen bacterial cells from complex environmental communities. The workflow featured three major phases: first, cells were separated from each other by encapsulating them in droplets during droplet generation. All droplets were collected in a vessel, where they lost their order. We refer to the state of a droplet population, in which the positional information of droplets is unrelated to the order of droplet generation, as 'droplets in bulk'. Second, droplets in bulk were incubated together using dynamic droplet incubation (*Mahler et al., 2015*), which ensured aerobic and homogenous cultivation conditions for months. Third, grown microbial cultures were isolated from droplets by depositing them on agar plates. Either all droplets were deposited in an untargeted manner or droplets were first screened for antimicrobial activity and then only selected droplets were distributed on agar plates (*Figure 1A*).

For designing the medium in which single cells were encapsulated in droplets, we followed the rationale of replicating the conditions in the natural environment as closely as possible. We used low amounts of defined nutrients typically originating from plants supplemented with a cold-extracted soil extract (CESE), prepared from the same soil that the community was derived from, and several solid soil particles (diameter <40 μm). CESE contained dissolvable macro- and micronutrients and other effectors, like signaling molecules. The number of cells occupying a droplet during an encapsulation event follows a Poisson distribution (*Collins et al., 2015*). As we aimed at one cell per droplet, we minimized the probability of several cells per droplet by adjusting the concentration of cells at inoculation to obtain not more than 30% occupied droplets ($\lambda = 0.4$, on average 0.4 cells/droplet). Nevertheless, the occurrence of several cells per droplet cannot be excluded due to a small statistical probability of 6.16% and biological reasons like cells adhering to the same soil particle or to each other. The droplets were stabilized by a biocompatible surfactant forming a monolayer at the aqueous/oil interphase. The level of miniaturization (~200 pL droplet volume), in conjunction with bulk incubation, permitted droplet populations comprising on average $9 \times 10^6$ droplets.

### In-droplet cultivation results in higher CFU concentration in comparison to conventional plating

To prove the capability of our droplet cultivation technique, we directly compared the cultivation outcome of droplets with standard plating for the extracted microbial community of a brown earth soil sample (eight replicates, *Figure 1—figure supplement 1*). The soil community was incubated with the same media composition at 20 ˚C for 28 days in both approaches.

During incubation, we monitored microbial growth inside droplets by brightfield microscopy imaging followed by counting occupied droplets (*Figure 1B*). After 28 days, the average occupation frequency of the eight populations amounted to 22.4%, which translates into $3.03 \times 10^6$ colony-forming units (CFUs)/mL (*Figure 1—figure supplement 2*). On plates, the colony count is equivalent to $1.3 \times 10^6$ CFU/mL of the soil community. The more than twofold higher concentration of CFUs found during in-droplet cultivation indicates that the singularization and incubation of cells in droplets enabled more cells to replicate.

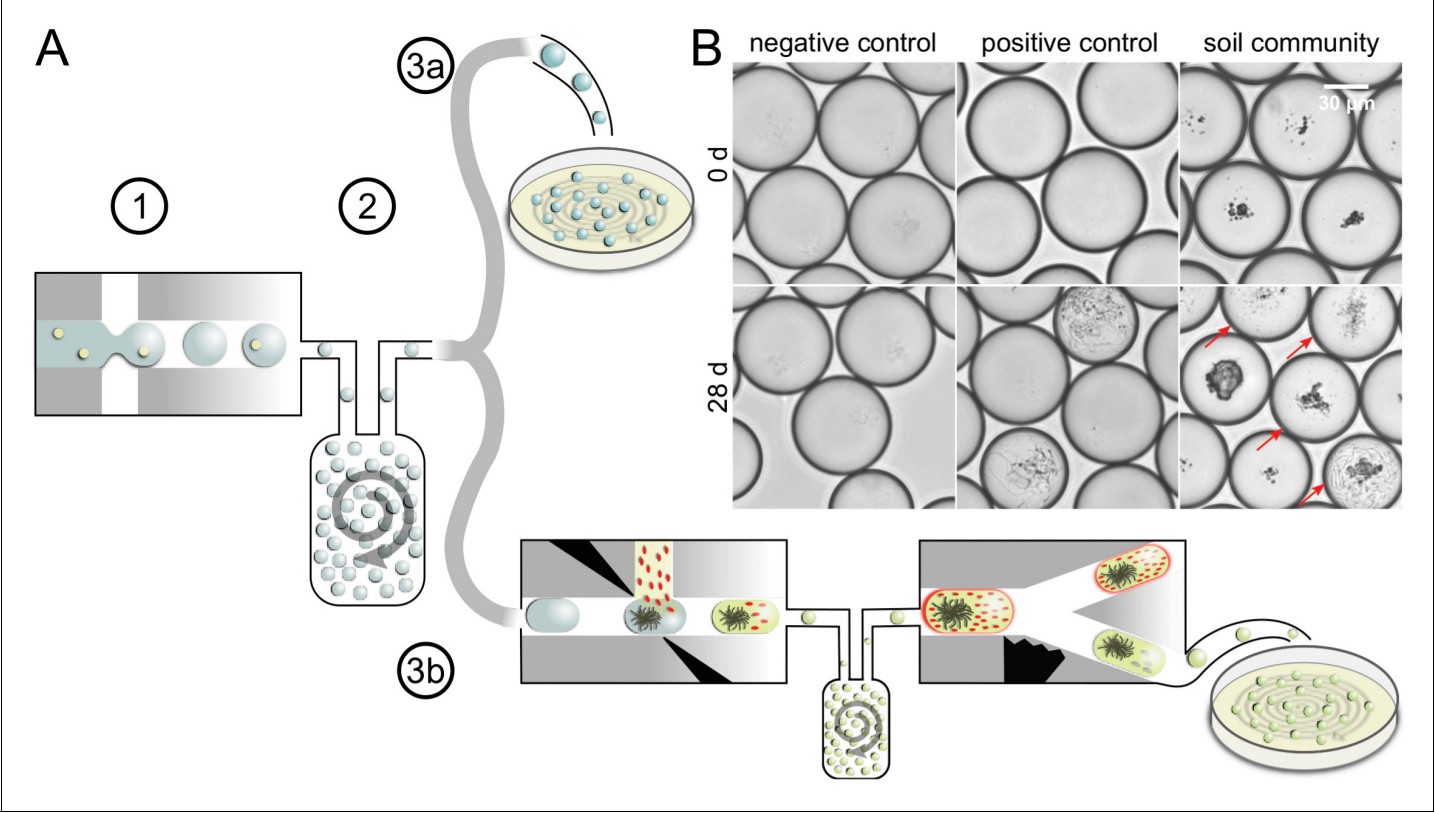

**Figure 1.** Concept of microbial droplet cultivation and activity screening. (**A**) General workflow of in-droplet cultivation with two options for strain isolation. (1) Single cells of the soil community are encapsulated in droplets. (2) Droplets are incubated under oxic conditions in bulk. (3a) Droplets are deposited on agar plates for strain isolation without previous selection. (3b) Droplets are screened for antibiotic producers and only screening hits are deposited for strain isolation. (**B**) Brightfield microscopy images of droplet samples after droplet generation and 28 days of incubation. Bacterial colonies were not visible right after droplet generation (top). Negative control droplets remained empty throughout the entire incubation period (bottom left). A mixture of five *Streptomyces* strains (*Table 1*) was used as a positive control (middle). Co-encapsulated soil particles are visible as dark particles in the images of droplets with soil community (right). A variety of different morphologies were observed in droplets with soil community, ranging from different sizes and densities of cocci- and rod-shaped cells to filamentous growth (bottom right). Droplets with cell colonies are marked with arrows.

The online version of this article includes the following figure supplement(s) for figure 1:

**Figure supplement 1.** Workflow for the comparison of cultivation outcome obtained with in-droplet and agar plate cultivation.

**Figure supplement 2.** Average colony-forming unit (CFU) concentration over time with different cultivation methods.

## A diverse set of bacterial species replicates in droplets

We analyzed the bacterial diversity of growing cells for each cultivation method by pooling the biomass of colonies after incubation, extracting metagenomic DNA, and sequencing the 16S rDNA amplicons on an Illumina MiSeq platform. We subsampled the reads at an equal sequencing depth per sample to avoid bias by variable library size (*Gihring et al., 2012*). Uninformative operational taxonomic units (OTUs) were removed by a combination of a total count and prevalence filter resulting in 2106 final OTUs.

To explore the similarity between the microbial community structures for the different cultivation approaches, we performed unconstrained ordination based on the community composition using Bray–Curtis distance (*Figure 2A*). Samples belonging to one sample type showed little dispersion, and plate and droplets-derived sample clusters did not overlap. These results indicate clear dissimilarities between the cultivation outcome of the two methods but a reproducible community composition within one cultivation method.

At the broad level of phyla, differences in the community structure are clearly visible between all sample types (*Figure 2B*). The relative abundance of the phylum Actinobacteria among the cultivated species was more than twofold higher in droplets (37.9%) than on agar plates (15.7%). As

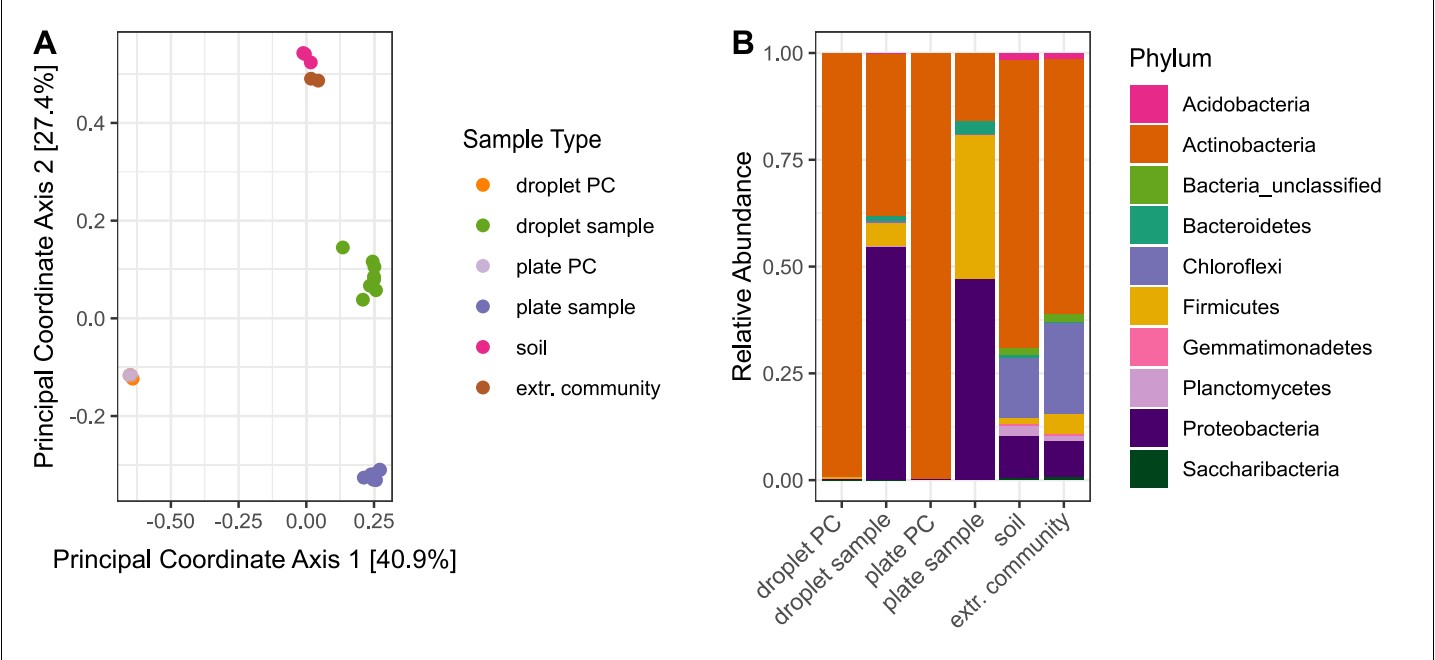

**Figure 2.** Differences in bacterial diversity due to cultivation technique. Sample-type abbreviations represent the following: droplet PC – droplet-positive control of five selected *Streptomyces* strains (*Table 1*); droplet sample – droplet population inoculated with soil community; plate PC – plate-positive control of five selected *Streptomyces* strains; plate sample – agar plates inoculated with soil community; soil – original soil sample; extr. community – extracted soil community. (A) Principal coordinate analysis based on Bray–Curtis distance matrix visualized similarities between microbial communities. (B) Community structure based on Illumina amplicon sequencing of the 16S rRNA gene. The phylogenetic affiliation at the phylum level is displayed for each sample type. All samples for one sample type were merged. The bars depict the relative abundance of the 10 most abundant phyla. The online version of this article includes the following source data, source code and figure supplement(s) for figure 2:

**Source code 1.** Source code for generation of both panels in *Figure 2*.

**Source data 1.** Source data for both panels in *Figure 2*.

**Figure supplement 1.** Alpha diversity indices for the subsampled 16S rDNA datasets with (A) total number of operational taxonimic units; (B) Total number of Singletons; (C) Good's Coverage; (D) Chao value; (E) Abundance-based Coverage Estimator (ACE); (F) Inverse Simpson's index; (G) Shannon index; and (H) Pielou's evenness.

expected, more than 99% of reads in the *Streptomyces*-positive controls (*Table 1*) affiliated to Actinobacteria for both cultivation techniques, proving the absence of external contaminations. The composition of the extracted soil community was similar to the initial community structure of the original soil sample, indicating no technical bias of extracting the microbial cells from the soil matrix.

Alpha diversity indices were calculated for the subsampled dataset without further filtering. The total number of OTUs, as well as Good's Coverage, chao1 (species richness) (*Chao et al., 2006*), and Shannon indices (species diversity) (*Shannon, 1948*) were significantly higher in the

**Table 1.** Bacterial species contained in the spore mixture constituting the positive control for the droplet–plate comparison.

| Species | ID |
| --- | --- |
| *Streptomyces griseus* | ST036300 |
| *Streptomyces hygroscopicus* | HKI0016 |
| *Streptomyces noursei* | JA03890 |
| *Streptomyces netropsis* | IMET43883 |
| *Streptomyces collinus* | IMET43780 |

droplet cultivation samples than in the plate cultivation samples, suggesting a higher microbial diversity cultivated in droplets (*Figure 2—figure supplement 1*).

To examine the differences in diversity in more detail, we directly compared the relative abundance of taxa on all taxonomic ranks for the cultivation outcome of droplets and plates (*Figure 3—figure supplements 1–5*). A pronounced distinction was the significantly higher abundance of the phylum Firmicutes on plates, comprising one-third of the community (*Figure 3—figure supplement 1*), while it was represented with only 5.3% in droplets. On plates, Firmicutes were mainly composed of five OTUs (three affiliating to *Bacillus* and two to *Paenibacillus*). As expected, *Bacillus* and

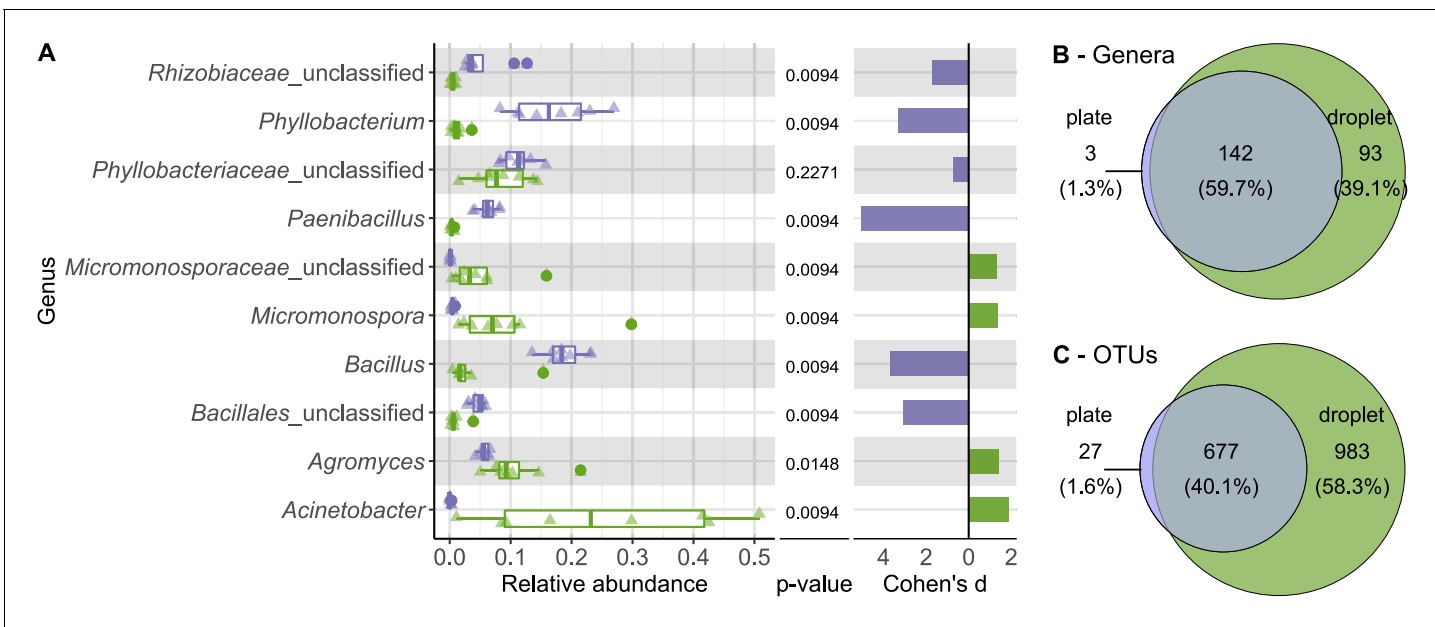

**Figure 3.** Comparing taxonomic classification on genus rank for replicating cells in agar plate and in-droplet cultivation. (**A**) Triangles depict the relative abundances for genera in eight biological replicates per droplet and plate cultivation method. Underlying boxplots show the distribution of data points. Relative abundances of operational taxonomic units (OTU) (evolutionary distance of 0.03) were agglomerated on the genus level (evolutionary distance of 0.1). Displayed are the 10 most abundant genera of in total 238 assigned genera covering 66.8% of all sequences. The 30 most abundant genera covering a total of almost 90% are shown in *Figure 3—figure supplement 5*. Means of relative abundances were compared by Wilcoxon rank-sum test for each genus (α = 0.05), applying Holm–Bonferroni correction for multiple comparisons. Adjusted p-values are displayed. As effect size, Cohen's *d* was computed and plotted as bars to indicate which differences are practically relevant. The direction and the color of the bars depend on the sample type in which the larger mean was found (blue – larger mean in plate samples, green – larger mean in droplet samples). (**B**) Venn diagram for the taxa on genus level. Of 238 assigned genera, 142 were found in both cultivation methods while 93 were unique to droplet cultivation and three were only observed in plate cultivation. The area of the Venn elements corresponds to the total number of genera found for the cultivation techniques. (**C**) Venn diagram for the OTU level. Of 1687 OTUs, 677 were found in both cultivation methods, 983 were unique to droplets, and 27 were unique to plates. The area of the Venn elements corresponds to the total number of OTUs found for the cultivation techniques.

The online version of this article includes the following source data, source code and figure supplement(s) for figure 3:

**Source code 1.** Source code for generation of (**A**) in *Figure 3*.
**Source code 2.** Source code generating *Figure 3—source data 2* and *3*.
**Source code 3.** Source code generating *Figure 3—source data 4* and *5*.
**Source data 1.** Source data for all panels in *Figure 3*.
**Source data 2.** Source data generated from *Figure 3—source data 1* containing all droplets genera for *Figure 3B*.
**Source data 3.** Source data generated from *Figure 3—source data 1* containing all plate genera for *Figure 3B*.
**Source data 4.** Source data generated from *Figure 3—source data 1* containing all droplet operational taxonomic units for *Figure 3C*.
**Source data 5.** Source data generated from *Figure 3—source data 1* containing all plate operational taxonomic units for *Figure 3C*.
**Figure supplement 1.** Comparing taxonomic classification on phylum rank for replicating cells in agar plate and in-droplet cultivation.
**Figure supplement 2.** Comparing taxonomic classification on class rank for replicating cells in agar plate and in-droplet cultivation.
**Figure supplement 3.** Comparing taxonomic classification on order rank for replicating cells in agar plate and in-droplet cultivation.
**Figure supplement 4.** Comparing taxonomic classification on family rank for replicating cells in agar plate and in-droplet cultivation.
**Figure supplement 5.** Comparing taxonomic classification on genus rank for replicating cells in agar plate and in-droplet cultivation.
**Figure supplement 6.** Detailed view into exclusive and uncultured taxa found in droplets or on plates.

*Paenibacillus*, together with *Phyllobacterium*, were the three most abundant genera on plates, accounting for 42% of all the sequences (*Figure 3A*). In contrast, *Acinetobacter* as a member of the γ-Proteobacteria showed the highest relative abundance on the genus rank for droplets, accompanied by *Agromyces* and *Micromonospora*. As a general trend, most taxa present on plates were also found in droplets, while many taxa were exclusively found in droplets but not on plates (*Figure 3B, C*, *Figure 3—figure supplement 6A*). Among the taxa that were exclusively found in droplet samples, which accounted for almost 60% of all taxa, we found also a considerable number of taxa classified as uncultured at the genus level (151 taxa of 983 exclusive droplet taxa) (*Figure 3—figure supplement 6B*). Taxa that were just unclassified and likely belong to the uncultured category were not considered for this comparison. While 15.2% of the exclusive droplet taxa were classified as uncultured, only 1.2% of the taxa found in both approaches – droplet and plate – were classified as uncultured and 0% in exclusive plate taxa.

## Microbial colonies can be isolated from microfluidic droplets

To facilitate the isolation of bacterial cells from droplets, we designed an experimental setup in which droplets were deposited on a nutrient containing matrix after 1 month incubation in order to enable microcultures to grow into macroscopically visible colonies (*Figure 4*). Droplets were injected into a capillary at 10–50 droplets/s and guided to an agar plate, mounted on a positioning system. While droplets and oil phase continuously emanated from the capillary tip, the plate was moved following a spiral pattern, leading to the distribution of droplets along the spiral streak over the agar plate. Both the volatile oil phase surrounding the droplets and, subsequently, also the aqueous droplets evaporated fast, leaving behind the cells that were previously confined in droplets now spatially separated on the agar surface.

We picked 224 colonies after incubation of the plates, trying to keep the selection random and thereby reflecting the true distribution of abundance for the bacterial diversity. In total, 301 axenic isolates were obtained on agar plates of which 266 were initially characterized by sequencing the full 16S rRNA gene. The isolates covered four major phyla resolving in 11 different genera (*Figure 5*). Most of the isolates were classified as *Bacillus* (52.3%), underlining the strong bias towards Firmicutes caused by intermediate cultivation on agar plates. Nevertheless, the accumulated abundance of Actinobacteria amounted to 30.5%, resembling the abundance in the droplet amplicon data. Moreover, the Actinobacteria were represented by several rarely observed genera like *Kocuria* and *Leifsonia*, and a notable high frequency of *Micromonospora* with 22.9%, which showcases the influence of in-droplet cultivation. Remarkably, one of the isolates clustered only to reference sequences of uncultured bacteria assigned to the family Cytophagaceae. However, this isolate could not be maintained for long on standard agar plates. The obtained genomic DNA could nevertheless be further investigated.

## Screening for antibiotic producers in droplets

We combined the cultivation of complex cell mixtures with an immediate screening for antibiotic compounds (*Mahler et al., 2018*) in order to achieve the selective isolation of putative producers of antimicrobial substances. For this purpose, we picoinjected cells of a reporter strain into all droplets of a population after 1 month incubation (*Video 1*). Based on the fluorescent signal, droplets with inhibited reporter cells (low red fluorescence signals) were selected in a droplet sorting operation on-chip (*Video 2*) and subsequently distributed on agar plates to recover soil bacteria-producing antimicrobial compounds.

In four different screening experiments, we explored combinations of two different reporter strains, *Escherichia coli* and *Bacillus subtilis,* as representatives of Gram-negative and Gram-positive bacteria, and two different media to cultivate the soil community in droplets (*Supplementary file 1*). We observed among the inhibiting isolates an enrichment of Actinobacteria and Alphaproteobacteria represented by various genera like *Kocuria*, *Microbacterium*, and *Sphingomonas* for the *B. subtilis* reporter (*Figure 6*). In contrast, the spectrum of genera recovered with the *E. coli* reporter was broader. The most abundant genus for both reporters was, however, *Bacillus sp.* . Interestingly, two isolates falling in the Cytophagaceae family but being only distantly related to known type strains (<95%) were obtained. They also did not cluster closely to the previously uncultivated Cytophagaceae isolate gained in the untargeted isolation (*Figure 6—figure supplement 1*), indicating that

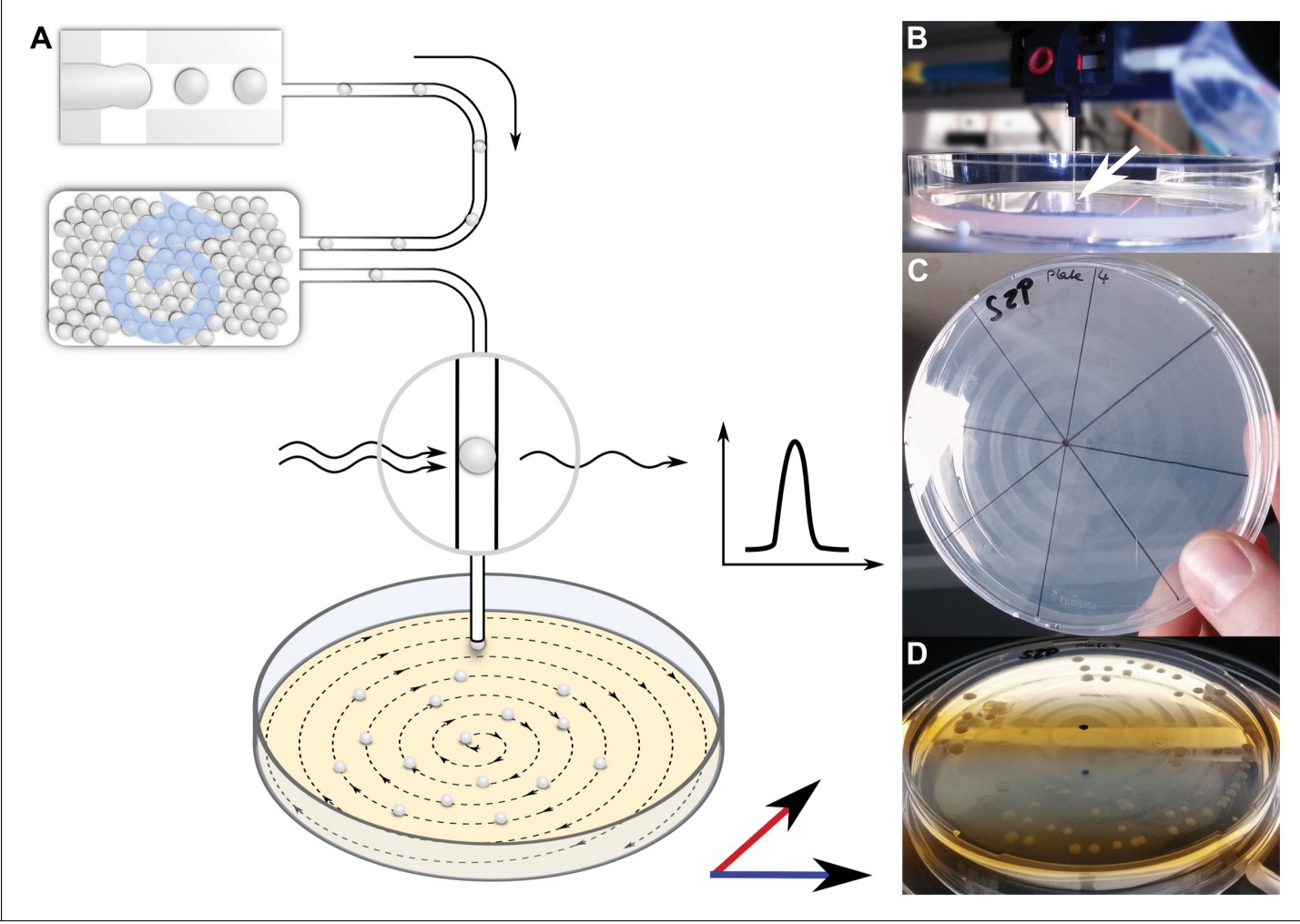

**Figure 4.** Continuous droplet deposition of preincubated droplets on an agar plate. (A) Sketch of continuous droplet deposition workflow. Droplets inoculated with cells of the soil community were incubated for 1 month before they were injected into a capillary (100 μm ID) positioned above an agar plate. The agar plate moved in x–y direction at a speed of 1 mm/s, while the droplets were injected at 10–50 droplets/s. The droplet frequency was monitored by a custom-developed sensor. (B) Image of the capillary taken while droplets are deposited on an agar plate. The tip of the capillary is highlighted by a white arrow. The capillary tip is close to the agar surface in order to establish a continuous fluid film. (C) An agar plate immediately after droplet deposition with the dried oil film in the spiral pattern of continuous deposition. (D) An agar plate with deposited droplets after incubation for 2 weeks. The dried oil film is still visible, on which the bacterial colonies are now aligned.

those isolates represent an additional new strain. As before, these isolates could not be maintained on agar plates for long.

## Identifying the chemical arsenal of a potent antibiotics producer strain

For detailed natural product discovery, we chose one of the droplet isolates that showed strong inhibition of various test strains like *Mycobacterium vaccae* or *Pseudomonas aeruginosa* during the primary validation of our screening hits. This isolate (D121-0906-b3-2-1) is closely related to *Bacillus tequilensis* (99.79%) and was isolated using *B. subtilis* as reporter strain in droplets. When challenged with *M. vaccae*, isolate D121-0906-b3-2-1 showed a peculiar increase in growth rate (*Figure 7—figure supplement 1*). Additionally, the isolates' crude culture supernatant exhibited a strong antifungal activity against test strains like *Candida albicans, Sporobolomyces salmonicolor,* and *Penicillium notatum* (*Supplementary file 1*—table 2). Other isolates with less remarkable initial inhibition properties are still under detailed investigation.

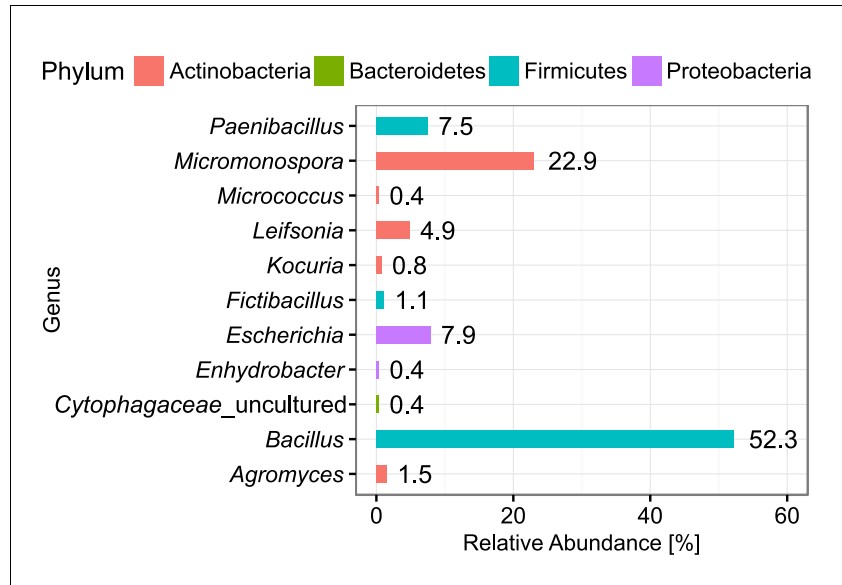

**Figure 5.** Abundance of classified bacterial isolates obtained from droplet cultivation. The distribution of abundance on the genus rank is displayed. Colors of the bars indicate the affiliation to phyla. Each bar is labeled with its exact relative abundance.

The online version of this article includes the following source data and source code for figure 5:

**Source code 1.** Source code for generation of *Figure 5*.

**Source data 1.** Source data for *Figure 5*.

For increasing the production titer of the natural products, we cultured *Bacillus* sp. D121-0906-b3-2-1 under several conditions including varying media composition and duration of cultivation (*Figure 7A*, *Figure 7—figure supplement 2*). In addition, extraction took place with either ethyl acetate or the absorber resin XAD-2. The extracts were analyzed by liquid chromatography coupled to high resolution mass spectrometry (LC-HRMS) and five compounds were detected. For compounds **1** and **2** with $m/z$ 579 [M–H]⁻ and $m/z$ 741 [M–H]⁻, respectively, we deduced the molecular formulas of $C_{34}H_{47}N_2O_6$ (**1**) and $C_{40}H_{57}N_2O_{11}$ (**2**). The predicted molecular formulas as well as a characteristic UV absorption spectrum indicated that **1** and **2** are identical with bacillaene A and B, known strong antibacterial polyketides active against a variety of strains (*Patel et al., 1995*; *Müller et al., 2014*; *Figure 7B*, *Figure 7—figure supplement 4A, B*). Tandem mass spectrometry (MS/MS) fragmentation analysis confirmed this

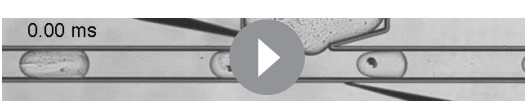

**Video 1.** Picoinjection of reporter cells to droplets inoculated and incubated with the soil community. Images were taken with 800 frames/s at 5x magnification in bright-field illumination. Images are played with 7 frames/s. In the field of view is the channel that is guiding the reinjected droplets through the electrical field applied between the black electrodes (only tips are visible). The surfactant-stabilized water oil interphase is destabilized by the electrical field resulting in fusion of the aqueous phase coming from the top channel and containing reporter cells and the aqueous droplet. When droplets leave the electrical field, the interphase is stable again.

https://elifesciences.org/articles/64774#video1

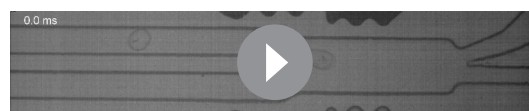

**Video 2.** Sorting of droplets based on their fluorescence intensity in the red channel. Images were taken with 1577 frames/s at 5x magnification. Every third image is played with 10 frames/s, resulting in a 52-fold reduction in playback speed. In the field of view is the droplet sorting structure. Droplets with high red fluorescence intensity are pulled with an electrical field into the upper channel. Droplets with low red fluorescence intensity leave the chip through the lower outlet.

https://elifesciences.org/articles/64774#video2

assumption. Cleavages of the glucopyranose unit in **2** as well as of the peptide bond (mass shift *m/z* –163 Da) in **1**–**2** were observed (*Figure 7—figure supplement 4E, F*). Furthermore, we detected three highly lipophilic metabolites (**3**–**5**) in the extracts of *Bacillus* sp. D121-0906-b3-2-1 with molecular masses of *m/z* 1006 [M–H]⁻ (**3**), *m/z* 1020 [M–H]⁻ (**4**), and *m/z* 1034 [M–H]⁻ (**5**), respectively. High-resolution MS data, as well as characteristic amino acid fragments obtained from HRESI-MS/MS analysis, revealed these cryptic molecules to be surfactins (**3**–**5**), cytotoxic lipopeptides (*Figure 7C*, *Figure 7—figure supplement 5D–F*; *Kakinuma et al., 1969*; *Bonmatin et al., 2003*). The retention times of a surfactin standard (Merck) were identical with **3** (*Figure 7—figure supplement 5A*). To exclude a linearized form of the surfactins, called gageostatins (*Tareq et al., 2014*), we added base (0.4 M NaOH) to the extracts to cleave the ester bond of **3**. A mass shift of +18 Da was observed, confirming a cyclized form of the lipopeptides (*Figure 7—figure supplement 5C*). Surfactins and surfactin-like natural products are known to exert a broad spectrum of antimicrobial effects against Gram-positive and Gram-negative bacteria as well as fungi (*Vitullo et al., 2012*; *Liu et al., 2012*; *Luo et al., 2015*; *Jiang et al., 2016b*) and may thus explain the strong antimicrobial effects of the *Bacillus* sp. extracts towards the test strain panel. We fractionated the crude extract with preparative high performance liquid chromatography (HPLC) and submitted the bacillaene- and surfactin-containing fractions to an agar diffusion assay against *B. subtilis* (*Figure 7—figure supplement 3*). The bacillaene fraction inhibited growth of *B. subtilis* most effectively while the surfactin fractions showed weaker antibacterial activities.

## Discussion

We devised an innovative microscale cultivation strategy utilizing a modular droplet-microfluidic platform and mimicking environmental conditions in situ. A miniaturized droplet volume of 200 pL facilitated high-density long-term cultivation of 2 million compartmentalized single cells in a total volume of 1.5 mL. Additionally, we were able to screen for antibiotic production, which demanded the addition of reporter strains to the cultivation compartments.

One major advantage of our method compared to previous microscale cultivation systems lies in the space-efficient incubation of droplets in bulk. Thereby, we attain a 180-fold higher number of droplets and a higher cultivation density than the microfluidic streak plate (*Jiang et al., 2016a*), for instance (50.000 × 180 pL droplets in an array). In contrast to systems relying on agarose-stabilized gel microdroplets incubated in aqueous medium (*Zengler et al., 2002*; *Akselband et al., 2006*), our method does not suffer from cross-contamination of cells between the compartments or compartment and continuous phase. Although we could not use commercial FACS instruments for droplet sorting, we demonstrated a stable on-chip sorting, which enabled our screening procedure. The screening was conducted after an extensive aerobic incubation period allowing environmental cells to replicate and reach the physiological state for natural product biosynthesis before the reporter strain was injected into the droplets. Such a protocol is not feasible when double emulsions (*Terekhov et al., 2017*) or gel-stabilized droplets are used.

For the cultivation of a complex bacterial community rich in recalcitrant and slow growing genera, we have demonstrated the advantages of our in-droplet cultivation strategy. A more diverse subset of cultured bacteria and a twofold higher CFU concentration compared to plates have presumably been achieved for three reasons: (i) absence of competition in single-cell confinement, (ii) the high number of incubated cells, and (iii) the new and unique cultivation conditions in our droplets during long-term dynamic droplet incubation.

We also specifically chose conditions like low nutrient amounts and long incubation times that might favor growth and metabolic activity of slow growing species over fast growing species. We installed this bias towards slow growing species in our screening as a dereplication strategy because fast growing species are usually readily culturable and have been extensively screened in standard natural product screenings. In future work, we even consider to actively deselect for fast growers by image-based droplet sorting within the first week to further reduce the chance of rediscovery of natural products.

It is noteworthy that the surfactant-stabilized droplet boundary, though forming a mechanical barrier to prevent cell cross-contamination, is not completely abolishing molecular transport for all types of molecules. In general, only a few molecules with relatively high partitioning coefficients, meaning fairly hydrophobic compounds, might leak into the oil (*Skhiri et al., 2012*). Other molecules, which

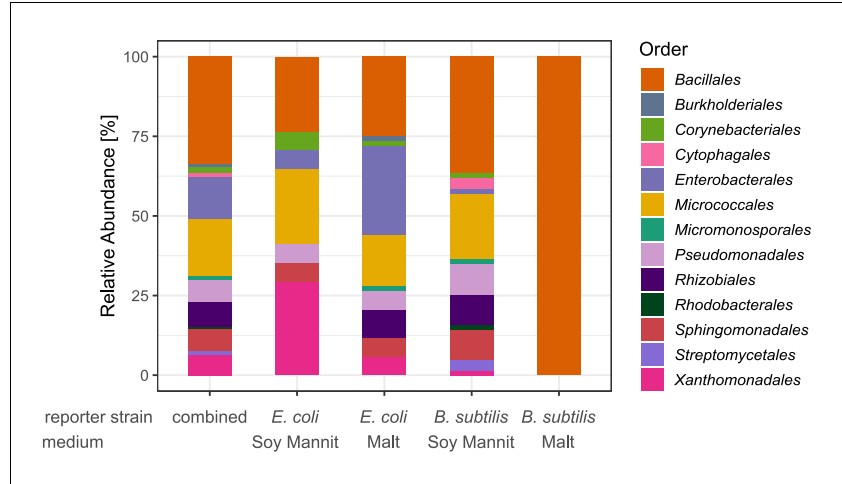

**Figure 6.** Taxonomic diversity of isolates obtained in the screening for antibiotic producers. The classification is depicted on the level of order and based on the Sanger sequences of the nearly complete 16S rRNA gene. The bars depict the relative abundance of all detected orders for the isolates of all four screening rounds combined and for each screening round separately.

The online version of this article includes the following source data, source code and figure supplement(s) for figure 6:

**Source code 1.** Source code for generation of *Figure 6*.
**Source data 1.** Source data for *Figure 6*.
**Figure supplement 1.** Maximum likelihood tree based on nearly full-length 16S rRNA gene sequence.

are usually poorly soluble in the perfluorinated continuous phase, can only partition into the continuous phase mediated by excessive surfactant molecules, unbound to the droplet border (*Gruner et al., 2016*). This kind of interdroplet transport can be minimized by decreasing surfactant concentrations, as was done here. Although the risk for chemical cross-talk is low, it could be problematic for the detection of antibiotics. Local concentrations of traveling compounds in producer droplets would be diluted towards other droplets causing false-negative detection. In contrast, chances for false-positives, that is, leakage of a specific compound from droplet with confined producer and its accumulation in one random other droplet without producing organism, are low due to constant droplet mixing and the comparably large amount of oil used during incubation (~7 mL oil vs. 200 pL droplet × 9000 = 1.8 µL, assuming that one in 1000 droplets contains a producers of the same leaking compound). Overall, the risk to miss antibiotics is higher for hydrophobic compounds, making the system favor the detection of hydrophilic substances. We understand the focus on hydrophilic antibiotics as a strength since the outer membrane of Gram-negative bacteria, including many clinically relevant pathogens, makes them more resistant to a number of hydrophobic antimicrobials (*Savage, 2001*). Interdroplet transport of molecules might also present an advantage in the cultivation phase. Allowing communication between species might preserve to some extent the growth-stimulating influence of multispecies communities.

We were able to recover part of the increased bacterial diversity during strain isolation from droplets, exemplified by several rare genera (e.g., *Kocuria* sp. and *Leifsonia* sp.) and a candidate for a new species in the family of Cytophagaceae, among relatively few isolates. Presumably, the yield of unusual isolates could be further increased by solely focusing on exceptional morphologies during colony picking. We avoided to do so to learn about the abundance distribution of species we obtained with the practiced isolation workflow. Our isolation procedure enabled us to spread droplets on agar plates without mechanically disrupting cell agglomerates. Hence, also droplets that contained large cell numbers were represented by only one colony on the plate. Nevertheless, the overall taxonomic distribution found after in-droplet cultivation could not be recovered with our droplet depositioning and isolation procedure on agar plates. The isolate collection was biased by organisms abundantly found on solid media plates and a number of species able to grow in droplets could not be recovered. The strongest impact might be through the change from liquid to solid

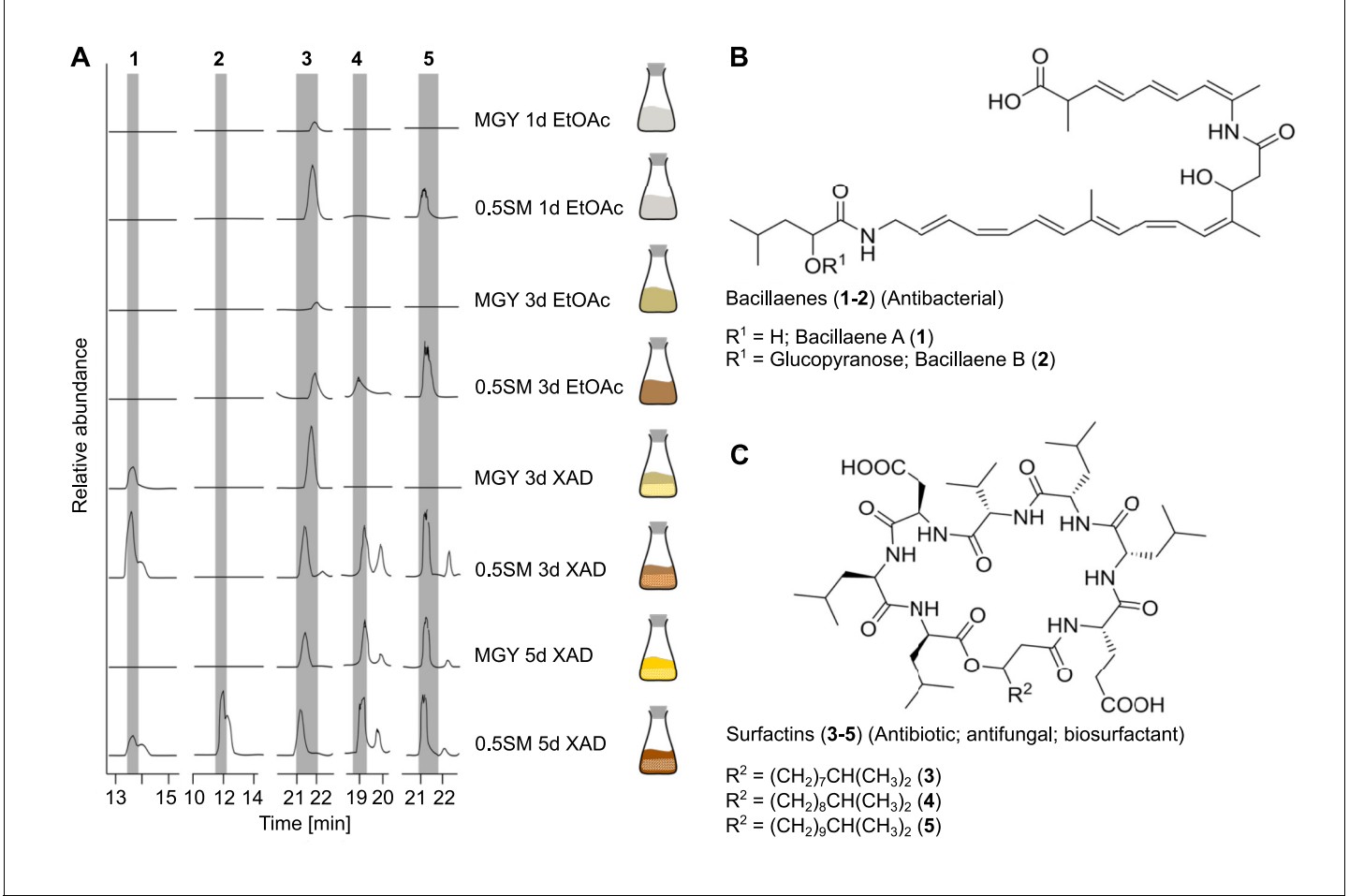

**Figure 7.** Detected natural product arsenal of *Bacillus* sp. strain D121-0906-b3-2-1. (**A**) Metabolic profiles of *Bacillus* sp. extracts under different growth and extraction conditions detected by liquid chromatography coupled to high resolution mass spectrometry (LC-HRMS) (displayed as extracted ion chromatograms in negative mode). (**B**) Bacillaene A (**1**) and B (**2**) and (**C**) surfactin-like compounds (**3–5**) detected in extracts of *Bacillus* sp. isolate D121-0906-b3-2-1. Tandem mass spectromety (MS/MS) spectra for **1–5** are provided in *Figure 7—figure supplements 4* and *5*.

The online version of this article includes the following figure supplement(s) for figure 7:

**Figure supplement 1.** Primary screening validation bioactivity assays with D121-0906-b3-2-1 crude culture supernatant and selected test strains.

**Figure supplement 2.** Metabolic profile of *Bacillus* sp. D121-0906-b3-2-1 (lower panels, soy mannitol medium + cold-extracted soil extract [CESE], 5 days cultivation, XAD adsorber extraction) in comparison to medium control (soy mannitol medium + CESE, XAD extraction).

**Figure supplement 3.** Bioactivity assay with diluted fractions of *Bacillus* sp. extract against *B. subtilis*.

**Figure supplement 4.** Characterization for bacillaene-like compounds (**1** and **2**) in extracts of *Bacillus* sp. isolate D121-0906-b3-2-1.

**Figure supplement 5.** Characterization for surfactin-like compounds (**3–5**) in extracts of *Bacillus* sp. isolate D121-0906-b3-2-1.

cultivation conditions. Large differences in culturability for several known strains and isolation campaigns have been described depending on the choice of liquid or solid cultivation media (*Nosho et al., 2018*; *Shigematsu et al., 2007*; *Wai et al., 2000*). Hence, it can be assumed that this is the case also for many so far unknown species. We, therefore, hypothesize that the change from liquid to solid cultivation contributed to the loss of some isolates, which could not be maintained in culture for extended periods outside of droplets. Consequently, we are currently developing other upscaling strategies into liquid culture like droplet deposition in microtiter plate wells. However, this bias likely does not abolish a droplet-aided strain discovery with the here presented droplet depositioning. We consider the theories of the scout model, postulating that cells of a dormant population switch stochastically into an active state scouting the current conditions, and spontaneous domestication as described by Buerger and Epstein (*Bunge and Epstein, 2009*; *Buerger et al., 2012a*; *Buerger et al., 2012b*). The high parallelization in droplet cultivation leads to substantially increased

chances to find rare scout cells, which can replicate and turn into culturable variants (spontaneous domestication). Those domesticated variants would subsequently also grow on agar plates and become accessible within our droplet workflow.

We increased the practical relevance of our cultivation approach by combining it with a screening step in order to detect and specifically select droplets containing environmental cells producing compounds with antibiotic activity. Even though axenic cultures of putative new species could not be maintained with standard microbiological cultivation after isolation from droplets, the cells could be used for obtaining high-quality genome information. Thereby, our method assists in strain prioritization during genome mining or guide future isolation campaigns. Additionally, cloning and heterologous expression of unusual biosynthetic gene clusters with non-canonical domain architectures might pave the way for identifying novel antibiotics that would have otherwise stayed undetected.

To further increase the number of rare taxa cultivated in droplets, we could charge the perfluorinated oil carrying dissolved gases with nitrogen or other gases during dynamic droplet incubation. Thereby, a microaerophilic or even anaerobic incubation would be enabled, which was recently shown to dramatically increase the species richness in droplets for human microbiomes (*Watterson et al., 2020*; *Villa et al., 2020*). We are also working on a less bulky droplet generation setup, which can be moved easily into an anaerobic chamber for anaerobic droplet generation. With that also obligate anaerobic bacteria like from the human or animal gut could be cultured. To conduct also the screening with bacterial reporter strains under anaerobic conditions, a more comprehensive adaptation of the screening system would be required, encompassing selection of anaerobic reporter strains and fluorescent proteins that do not need oxygen for maturation. Besides encapsulating single cells, several cells could be confined per droplet creating simplified multispecies communities, which have the potential to either support growth of dependent species (*Tanaka and Benno, 2015*) or to exert a change in natural product biosynthesis (*Shi et al., 2017*; *Molloy and Hertweck, 2017*). Furthermore, we here have isolated strains after one incubation round in droplets, but alternatively the emulsion can be broken under controlled conditions and a new droplet cultivation can be started from the pooled cell suspension. By passaging a community several times through the in-droplet cultivation, replicating organisms become enriched, and the chance of strain domestication increases (*Bollmann et al., 2007*).

Our cultivation and screening strategy has proven to be comprehensive and adaptable to the challenging needs of fastidious microorganisms representing interesting sources for natural products. Therefore, we envision that the bacterial diversity of numerous natural habitats, especially including environments with limited access to samples (e.g., patient biopsies), will be explored in more depth with our strategy. The combination of bacterial strain discovery during the in-droplet cultivation phase with subsequent selection of the most auspicious candidates for natural product biosynthesis through a highly robust reporter assay will streamline our efforts towards new scaffold structures for the development of antibiotics.

## Materials and methods

### Microfluidic device fabrication and operation

Microfluidic chips were designed in AutoCAD 2015 (Autodesk Corp., USA). Designs were printed by JD phototools (Great Britain) as photomasks, which were used by Biotec TU Dresden Microstructure facility to fabricate SU8 molds. PDMS (Sylgard 184, Dow Corning, Germany) replicas were obtained from molds following standard protocols (*Mazutis et al., 2013*) and plasma bonded to microscope glass slides. For hydrophobization, Novec 1720 (3M, Germany) was introduced into the channels while heating the chips to 100℃. At the same temperature, channels designated for electrodes were filled with low-melting solder (Indalloy 19, Indium Corporation of America, USA). Fluids were actuated by high-precision syringe pumps (neMESYS, Cetoni GmbH, Germany) and pressure pumps (MFCS-EZ, Fluigent, France). PTFE tubing (1/16″ OD, 0.25 mm ID and 0.5 mm ID, Chromophor Technologies, Germany) was used to realize fluidic connections. The device for droplet incubation was 3D-printed by i.materialise (Belgium) (the CAD design files are included in *Supplementary files 2* and *3*), and the complete setup was used as described before (*Mahler et al., 2015*) in a humidified chamber. Microfluidic operations were conducted on an inverted microscope (Axio Observer Z1, Zeiss, Germany) and imaged during flow with a Pike F-032B ASG16 camera (Allied Vision

Technologies, Germany). For images of stationary droplets in observation chambers, a pco.edge 5.5 m camera (PCO AG, Germany) was employed. Chips containing observation chambers were also manufactured from PDMS bonded to glass and comprised a wide channel (2 mm wall-to-wall, 150 µm height), termed chamber. Droplets with oil were introduced into the chamber until all air was replaced and entrances to the chip were closed. After 3–5 min, droplets stopped moving and spread into a monolayer ready for imaging by transmitted light microscopy. Droplets were constantly surrounded by oil, thereby maintaining their integrity.

## Soil sampling and extraction

Soil was collected from a dry grassland site at the nature reserve 'Schwellenburg' (51° 1 '519174' 'latitude', 10° 57 '146013' 'longitude', 263 m altitude) close to Erfurt (Germany) in May 2016. Soil was collected from a depth of 10 cm after the overlaying turf was removed. It was transported in a sealed plastic bag for immediate further processing in the lab.

For the CESE, the soil was mixed with aq. dest. in a ratio of 1:1 (w/w). At room temperature, the mixture was stirred for 2 hr at 300 rpm and then left for sedimentation for 2 hr. Insoluble particles in the decanted supernatant were removed by centrifugation at 15,970 × g for 20 min. The resulting supernatant was vacuum-filtered twice through cotton wool. For final sterilization, the filtrate was sterile-filtered with bottle-top membrane filters (pore size 0.2 µm) and subsequently aliquoted and stored at –20 °C.

To extract the bacterial soil community, 20 g of the freshly sampled soil spread onto a flat glass tray were dried for 24 hr at 37 °C. The dried soil was ground in a mortar, while stones and macroscopic plant material were removed. Ten grams of the milled soil and 25 glass beads (diameter 2–6 mm) were suspended in 90 mL CESE and shaken at 200 rpm for 2 hr at 28 °C. Subsequently, the suspension was sonicated twice at 50 W for 1 min. One half of the suspension was vacuum-filtered through a cell strainer (40 µm pore size). The other half was left for particle sedimentation at room temperature. After 1 hr, the supernatant was decanted. Aliquots of both cell suspensions were stored at –20 °C. Upon usage, the suspension rich in soil particles (decanted) was mixed with the suspension containing fewer particles (filtered) in a ratio of 2:35 (v/v).

## Microorganisms and culture conditions

The soil community was cultivated in medium containing 50% (v/v) CESE, 6% (v/v) supernatant of soy mannitol medium and 44% (v/v) deionized water (*Supplementary file 1*—table 3). For agar plates, 2% (w/v) agar was autoclaved with appropriate amounts of water and soy mannitol supernatant at 121 °C for 20 min. After cooling to 56 °C, CESE was added and plates were poured. For cultivation on plates, the soil community was diluted with 0.9% (w/v) NaCl solution in a serial dilution until $10^{-4}$; 50 µL of the respective dilution were spread on agar plates with Drigalski spatulas. For droplet generation, the soil community was suspended in medium without dilution. The positive control comprised a spore mixture of known *Streptomyces* strains (*Supplementary file 1*) in equal concentrations ($8 \times 10^5$ spores/mL) in the medium described above. All samples containing the soil community and respective controls were incubated at 20 °C.

For the inhibition assays in droplets, the reporter strains *E. coli* (*E. coli* JW1982 [*Baba et al., 2006*] with plasmid pMPAG6 – $P_{T5/lacO}$:mCherry) and *B. subtilis* (*B. subtilis* 168 *amyE::hy-mKATE:CM* [*van Gestel et al., 2014*]) were used. They were cultivated in 1× terrific broth (*Supplementary file 1*—table 4) containing 100 µg/mL ampicillin (*E. coli*) or 5 µg/mL chloramphenicol (*B. subtilis*) at 37 °C, 200 rpm for 16 hr. New cultures were inoculated to a start $OD_{600}$ of 0.1 and cultivated without selection markers until mid-exponential phase. Before picoinjection of reporter cells into droplets, cells were pelleted and resuspended in 2.5× terrific broth + 1% (w/v) glucose (*Supplementary file 1*—table 4) to a final $OD_{600}$ of 4. In the case of *E. coli*, 0.5 mM isopropyl β- d-1-thiogalactopyranoside (IPTG) was added.

## Comparison of cultivation in droplets and on agar plates

Microfluidic droplets were generated at a flow focusing unit using Novec HFE7500 (3M, Germany) with 0.5% Pico-Surf 1 (Dolomite, UK) as continuous phase. Droplets were generated at 1500 Hz with a volume of 200 pL. One droplet population, which was incubated in one droplet incubator, comprised approximately $9 \times 10^6$ droplets. Droplet populations were generated with eight replicates

and compared to eight plate sets. One plate set consisted of eight plates with 1:100 dilution of the soil community and seven plates with 1:1000 dilution. The conducted replicates are regarded as biological since independent cultivation experiments were started from one common soil cell extract. The chosen sample size represents a compromise between maximum available droplet incubation units and minimum required replicate number for maximum statistical power.

For each cultivation technique, three positive controls were prepared and incubated along with the soil community samples. After an incubation time of 28 days, microcultures of one droplet population were pooled by fusing the emulsion with 200 µL/mL 1$H$,1$H$,2$H$,2$H$-perfluoro-1-octanol (Sigma, USA). The upper aqueous phase including the cells was transferred into a new vessel, and the cells were pelleted. From plates, colonies were pooled by suspending the cells in 2 mL 0.9% NaCl solution per plate with a cell scraper.

## DNA extraction and purification

For DNA extraction, 150 mg of sterilized glass beads (0.1–0.5 mm diameter) were added along with one volume of lysis buffer (100 mM Tris-HCl [pH 8], 20 mM Na-EDTA, 1.5 M NaCl, 1.2% [$w/v$] Triton X-100) to the cell pellets. In case of cell pellets derived from plates, the double amount of glass beads and three ceramic beads (0.5 cm diameter) were used. Cells were lysed in a FastPrep-24 (MP Biomedicals, Germany) two times for 2 min at a speed of 6 m/s. Afterwards, 0.3 volumes of 20% (w/v) SDS (Roth, Germany) were added, and precipitates removed by centrifugation. The cell lysate was extracted with one volume of phenol:chloroform:isoamylalcohol (25:24:1, Roth, Germany). To remove phenol traces, the samples were further extracted three times with one volume of chloroform:isoamylalcohol (24:1, Roth, Germany). Finally, the DNA was precipitated with ethanol. DNA samples were mixed with 0.1 volume 3 M sodium acetate, 0.001 volume of 20 mg/mL glycogen, and 2.5 volumes of 98% (v/v) cold ethanol. Samples were stored at −20 ˚C for at least 16 hr before they were centrifuged for 1 hr at 4 ˚C and 25,000 × $g$. The supernatant was removed and the remaining pellet washed with 1 mL 70% (v/v) ethanol. The samples were centrifuged and decanted again. Traces of ethanol were removed by drying the pellet for 5 min under vacuum. Afterwards, the DNA was rehydrated in NE Buffer (5 mM Tris/HCl, pH 8.5) and stored at 4 ˚C.

## Further purification of metagenomic DNA via Q Sepharose columns

To remove humic acids, the DNA was further purified with custom-made Q Sepharose columns as described by *Mettel et al., 2010*. For each DNA sample, 600 µL of Q Sepharose (Q Sepharose High Performance, GE Healthcare, USA) were aliquoted in 1.5 mL reaction vessels and washed four times in 1 mM potassium phosphate buffer (136 mg/L KH$_2$PO$_4$, 228 mg/L K$_2$HPO$_4$ × 3 H$_2$O). After washing, the Q Sepharose pellet was resuspended in 300 µL of the same buffer and transferred into a centrifugal filter unit (Durapore-PVDF 0.45 µm, Merck Millipore Ltd, Germany). The self-made column was packed by centrifugation, and the buffer that passed through the filter membrane was discarded. In the next step, the DNA sample dissolved in NE Buffer was pipetted onto the column and centrifuged for 10 s at 3000 × $g$, thereby DNA and humic acids were binding to the Q Sepharose. The DNA was eluted by adding 80 µL of a 1.5 M NaCl solution, which was also pushed through the column by centrifugation at 3000 × $g$ for 10 s. The elution step was repeated until the clearly visible brown band of humic acids reached the filter membrane, which usually happened after four repetitions. The DNA in the eluate was desalted by binding it to a silica membrane using the DNA Clean and Concentrator Kit (Zymo Research, Germany) following the manufacturer's protocol.

## DNase I pretreatment of polymerases

The polymerases were pretreated with DNase I (New England Biolabs, Germany) according to *Stach et al., 2001* in order to remove bacterial DNA traces that originated from the enzyme production processes. The necessary amount of polymerase stock was diluted in a ratio of 1:6 in PCR grade water. A 10× DNase I buffer and DNase I were added according to the manufacturer's protocol. The mixture was incubated 10 min at 37 ˚C and inactivated 10 min at 75 ˚C. After cooling down, the mixture was added to the master mix, which was subsequently aliquoted for the following PCR.

## 16S rDNA amplicon preparation and sequencing

The V3–V4 region of the 16S rRNA gene was amplified using the primers 314F and 758R (*Klindworth et al., 2013*) with Illumina adapters at the 5′ end. A Q5 polymerase (New England Biolabs, Germany) was used in a 50 µL reaction after pretreatment with DNase I. PCR was performed with the following settings: pre-denaturation at 98 ℃ for 3 min, 30 cycles of denaturation at 98 ℃ for 30 s, annealing at 55 ℃ for 40 s, elongation at 72 ℃ for 1 min, and final elongation at 72 ℃ for 5 min. DNA libraries were constructed from PCR products following the standard Illumina protocol. Amplicons were sequenced on an Illumina MiSeq system with 200 bp paired-end by IIT Biotech (Germany).

## Data analysis of sequence data

Forward and reverse reads of the amplicon data were merged by FLASH (*Magoc and Salzberg, 2011*). Primers were removed with cutadapt (*Martin, 2011*) allowing 20% error. For quality trimming, sickle (*Joshi and Fass, 2017*) was used with a minimum quality threshold of 20. Reads were further processed following the MiSeq SOP (https://www.mothur.org/wiki/MiSeq_SOP) in Mothur (*Schloss et al., 2009*). To detect chimera, the implemented uchime algorithm was used in combination with the SILVA reference database (release 128) (*Glöckner et al., 2017*), which was also used to taxonomically classify the reads. For clustering into OTUs, the vsearch method 'abundance-based greedy clustering' and a distance cutoff of 0.03 were employed. Read numbers across all samples were subsampled in Mothur to 88,435 sequences to obtain better comparability. No sample outliers were detected or removed. The OTU table was further processed and visualized in R and with the R package phyloseq (*McMurdie and Holmes, 2013*). OTUs were excluded when they were not represented more than three times in at least 10% of the samples. Furthermore, a median normalization within samples was applied to the read counts. Indices for alpha diversity are shown in supplemental information for subsampled data.

## Cell isolation from microfluidic droplets

Droplets were generated and incubated with the soil community as described above. After 1 month, droplets were reinjected from the incubator into a microfluidic chip and spaced to a frequency of 10–50 Hz. All droplets were directed to a glass capillary (TSH, 100 µm ID, 360 µm OD, Molex, USA), which led to a positioning system (neMAXYS 200, cetoni GmbH, Germany). The tip of the capillary was attached to a z-steerable arm of the system, while an agar plate was positioned below on an x–y-steerable platform. Thereby, droplets were continuously deposited in a spiral pattern on an agar plate over which the outlet of the capillary was moved at 1 mm/s. Plates were incubated for 15 days at 20 ℃, after which colonies were picked and streaked onto new agar plates. By restreaking multiple times, pure isolates were obtained of which DNA was extracted using the QIAamp DNA Mini Kit (QIAGEN, Germany).

## 16S rDNA amplification and analysis for isolate characterization

For isolate characterization, the whole 16S rRNA gene was amplified in 50 µL reaction using the primers 27F (AGA GTT TGA TCM TGG CTC AG) and 1492R (CGG TTA CCT TGT TAC GAC TT) and the PrimeSTAR GXL polymerase (Takara Bio, USA). Before amplification, the polymerase was treated with DNase I. The PCR was carried out as follows: pre-denaturation at 98 ℃ for 30 s, 35 cycles of denaturation at 98 ℃ for 20 s, annealing at 55 ℃ for 40 s, elongation at 68 ℃ for 30 s, and final elongation at 68 ℃ for 3 min. The Sanger sequencing was performed bidirectionally. Consensus 16S rRNA gene sequences of axenic cultures were assembled from forward and reverse reads with SeqTrace (*Stucky, 2012*) (v0.9.0) applying a Needleman–Wunsch alignment algorithm and a quality cutoff for base calls of 30. After automatic trimming until 20 consecutive bases were correctly called, consensus sequences were examined and curated manually. Consensus sequences of the nearly full-length 16S rRNA gene were aligned with SILVA Incremental Aligner (SINA) (*Pruesse et al., 2012*) (v1.2.11). Phylogenetic relations were deduced by reconstructing phylogenetic trees with ARB (*Ludwig et al., 2004*) (v6.0.6) using the 'All species living tree project' database (*Yarza et al., 2008*) (release LTPs128, February 2017). Sequences were added into the LTP-type strain reference tree using ARB parsimony (Quick add marked), and alignment was corrected manually. Phylogenetic tree calculation with all family members was based on a maximum-likelihood algorithm using RAxML

(*Stamatakis, 2006*) (v7.04) with GTR-GAMMA and rapid bootstrap analysis, a maximum-parsimony method using DNAPARS (*Felsenstein, 2005*) (v3.6), and neighbor-joining with Jukes–Cantor correction.

## Screening for antimicrobial compounds in droplets

Droplets inoculated with the soil community and incubated for 28 days were reinjected into a microfluidic chip at 200 Hz. The chip contained a picoinjection structure similar to the structures described by *Abate et al., 2010*. By applying an alternating electrical field through a function generator (AFG-2005, GW Instek, China) and a high-voltage amplifier (model 2210-CE, Trek, USA) with an amplitude of 20 $V_{pp}$, a frequency of 20 kHz, and a duty cycle of 50%, a cell suspension containing the reporter strain was picoinjected into each droplet. Droplets were subsequently guided into a new incubator, and soil organisms and reporter cells were co-incubated for 20 hr at 28 °C inside droplets by applying dynamic droplet incubation. Finally, droplets were reinjected into a droplet sorting structure (*Baret et al., 2009*), where fluorescent proteins of the reporter strains were excited with a laser (488 nm diode laser, LASOS, Germany) and emitted light detected with a photomultiplier module (H10721-20, Hamamatsu Photonics UK Limited, UK). Droplets exceeding a predefined intensity threshold were sorted via dielectrophoresis by a triggered burst signal of 430 V, 30 cycles, 6 kHz, 50% duty cycle. Software for data logging was custom-made and written in LabVIEW (v2015). Sorted droplets with high fluorescence intensity were discarded while the remaining droplets were collected and distributed on agar plates. Plates were incubated, and colonies were picked and purified.

For primary validation of hits, selected isolates were cultivated in 50 mL of various media (*Supplementary file 1*—table 5) at 28 °C, 160 rpm, for 3–10 days. Crude culture supernatants were obtained by centrifugation at $20,000 \times g$ for 15 min at 20 °C and stored at −20 °C until testing. Antimicrobial activity was characterized in agar diffusion tests against a primary panel of test strains, which included different species of Gram-positive (*B. subtilis* ATCC6633, *Staphylococcus aureus* SG511, *M. vaccae* IMET10670) and Gram-negative bacteria (*E. coli* SG458, *P. aeruginosa* SG137) as well as fungal strains *S. salmonicolor* SBUG549, *C. albicans* ST50163, and *P. notatum* JP36. The antibiotics ciprofloxacin 5 μg/mL and amphothericin B 10 μg/mL were used as positive reference for bacterial and fungal test strains, respectively. For bacterial test strains, 34 mL standard 1 nutrient agar (Merck 1.07881) were inoculated with 100 μL of bacterial suspension adjusted to an optical density of McFarland standard 0.5 (for *M. vaccae* IMET10670 McFarland standard 1 [Biomerieux 70900], Inoculum 200 μL). For agar diffusion tests with fungal strains JP36 and SBUG549 malt extract agar (Oxoid CM 59) and for ST50163 Yeast-Morphology-Agar (Difco 239320) were inoculated with cell suspensions, resulting in a final concentration of $5 \times 10^6$ cells/mL. The inoculated agar media were immediately poured into leveled test plates, and after solidification 12 holes were punched. Then, 50 μL of crude supernatants, reference antibiotics, or sterile culture media were added per hole. After incubation for 18 hr at 37°C (bacterial tests) and 30°C (fungal tests), the diameter of growth inhibition zones surrounding the holes was measured manually (in mm) (*Supplementary file 1*—table 2).

## Secondary metabolite production

For the identification of natural products, *Bacillus* sp. strain D121-0906-b3-2-1 was inoculated in diverse media (MGY+M9 [*Dose et al., 2018*] and soy mannitol medium + CESE [SM]) at $OD_{600}$ 0.1 and grown at 160 rpm, 28 °C for 3–8 days. Extraction took place with 1:1 volume of ethyl acetate overnight or with addition of XAD-2 to the culture broth for 30 min followed by elution twice with 100% methanol for 30 min. The organic phase was dried with anhydrous sodium sulfate and concentrated under reduced pressure. Residues were dissolved in a small volume of methanol and measured at LC/MS. Fragmentation patterns were monitored via tandem mass spectrometry (MS/MS).

### LC/MS

Exactive Orbitrap High Performance Benchtop LC-MS (Thermo Fisher Scientific) with an electron spray ion source and an Accela HPLC System, C18 column (Betasil C18, 150 × 2.1 mm, Thermo Fisher Scientific), solvents: acetonitrile and distilled water (both supplemented with 0.1% formic acid), flow rate: 0.2 mL/min; program: hold 1 min at 5% acetonitrile, 1–16 min 5–99% acetonitrile,

hold 15 min 99% acetonitrile, 19–20 min 99% to 5% acetonitrile, hold 11 min at 5% acetonitrile. The metabolic profiles of the acquired fractions were monitored with HR-ESI-LC/MS.

## MS/MS

QExactive Orbitrap High Performance Benchtop LC-MS (Thermo Fisher Scientific) with an electrospray ion source and an Accela HPLC System, C18 column (Accucore C18 2.6 µm, 100 × 2.1 mm, Thermo Fisher Scientific), solvents: acetonitrile and distilled water (both supplemented with 0.1% formic acid), flow rate: 0.2 mL/min; gradient: hold 1 min at 5% acetonitrile, 1–10 min 5–98% acetonitrile, hold 12 min 98% acetonitrile, 22–22.1 min 98% to 5% acetonitrile, hold 7 min at 5% acetonitrile.

## Antimicrobial testing

A 400 mL PDB culture of *Bacillus* sp. strain D121-0906-b3-2-1 was grown at 30 °C and 120 rpm for 4 days. Absorber resin XAD-2 was added to the culture for 1 hr and, subsequently, extracted twice with 100% methanol for 1 hr. The organic phase was concentrated under reduced pressure. The crude extract was fractionated by preparative HPLC (Nucleodur C18 HTec, 5 µm VP 250 × 10 mm; flow rate: 5 mL/min; solvents: water supplemented with 0.01% trifluoroacetic acid and acetonitrile; gradient: 0–5 min 25% acetonitrile, 5–25 min 25–100% acetonitrile, 25–30 min 100% acetonitrile). Fractions were collected every 2–3 min, concentrated under reduced pressure, and the remains dissolved in methanol. The agar diffusion assays against *B. subtilis* 6633 B1 were conducted as previously described (*He et al., 2004*; *Ziehl et al., 2005*).

## Acknowledgements

We thank Karin Burmeister for excellent technical assistance and Dr. Àkos Kovács for kindly providing the *B. subtilis* reporter strain. We also thank Andrea Perner for LC/MS measurements and Christiane Weigel for assistance in antimicrobial testing. This work has been supported by the Thuringian Ministry of Education, Science and Culture and the European Fond for Structural Development (project MicroInfra no. 13008-715, CCI-Code 2007DE161PO001), the Thuringian Ministry of Economy, Labor and Technology (project DropCode, no. 2014FE9037), the German Center for Infection Research funded by the Federal Ministry of Education and Research (project no. TTU 09.811), the Jena School for Microbial Communication (JSMC) funded by the German Excellence Initiative, and the Deutsche Forschungsgemeinschaft (DFG, German Research Foundation, project no. 239748522 - SFB 1127).

## Additional information

### Funding

| Funder | Grant reference number | Author |
|---|---|---|
| Thuringian Ministry of Education, Science and Culture | 13008-715 | Lisa Mahler<br>Karin Martin<br>Martin Roth |
| Thuringian Ministry of Economy, Labour and Technology | 2014FE9037 | Lisa Mahler<br>Thomas Weber<br>Martin Roth |
| German Center for Infection Research | TTU 09.811 | Lisa Mahler<br>Karin Martin |
| Deutsche Forschungsgemeinschaft | GSC 214 | Lisa Mahler |
| Deutsche Forschungsgemeinschaft | 239748522 - SFB 1127 | Christian Hertweck<br>Sarah P Niehs<br>Kirstin Scherlach |

The funders had no role in study design, data collection and interpretation, or the decision to submit the work for publication.

## Author contributions
Lisa Mahler, Conceptualization, Formal analysis, Investigation, Methodology, Writing - original draft; Sarah P Niehs, Formal analysis, Investigation, Methodology, Writing - original draft; Karin Martin, Conceptualization, Formal analysis, Investigation, Methodology, Writing - review and editing; Thomas Weber, Formal analysis, Investigation, Writing - review and editing; Kirstin Scherlach, Formal analysis, Methodology, Writing - review and editing; Christian Hertweck, Supervision, Methodology, Writing - review and editing; Martin Roth, Conceptualization, Formal analysis, Supervision, Methodology, Writing - review and editing; Miriam A Rosenbaum, Formal analysis, Supervision, Writing - review and editing

## Author ORCIDs
Lisa Mahler ⓘ https://orcid.org/0000-0002-8241-9782
Sarah P Niehs ⓘ https://orcid.org/0000-0002-2012-5949
Thomas Weber ⓘ https://orcid.org/0000-0001-5048-0653
Christian Hertweck ⓘ https://orcid.org/0000-0002-0367-337X
Miriam A Rosenbaum ⓘ https://orcid.org/0000-0002-4566-8624

## Decision letter and Author response
Decision letter https://doi.org/10.7554/eLife.64774.sa1
Author response https://doi.org/10.7554/eLife.64774.sa2

## Additional files
### Supplementary files
- Supplementary file 1. Supplementary tables.
- Supplementary file 2. CAD design file of droplet incubator.
- Supplementary file 3. CAD design file of droplet incubator ferrule.
- Transparent reporting form

### Data availability
Amplicon sequence data were deposited to NCBI under the BioProject accession numbers PRJNA623865. For isolated axenic strains, 16S rRNA gene sequences were deposited to GenBank under the accession numbers MT320111 - MT320533. - Source data, encompassing numerical and or taxonomical data and R-analysis files, are provided as RData objects and R scripts for graphs: Figure 2ab, Figure 3abc, Figure 5, Figure 6. Figures 1ab, 4 and 7 do not contain analyzed experimental data but depict workflows and overview information.

The following dataset was generated:

| Author(s) | Year | Dataset title | Dataset URL | Database and Identifier |
|---|---|---|---|---|
| Mahler L, Martin K, Roth M, Rosenbaum M | 2020 | Amplicon studie of microbial diversity after in-droplet cultivation | http://www.ncbi.nlm.nih.gov/bioproject/?term=PRJNA623865 | NCBI BioProject, PRJNA623865 |

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
