## [Decision Letter]

**Acceptance summary:**

With the increasing incidence and spread of antibiotic resistant strains, discovery of new antibiotics is urgently needed. While most of the antibiotics currently in use were discovered from rather readily culturable soil microorganisms, there exists a huge non-culturable microbiome diversity in nature. This manuscript describes a potentially useful microfluidic platform that simplifies isolation and culture of single microbial species from complex environmental samples and to screen them for novel secreted antimicrobial molecules.

**Decision letter after peer review:**

Thank you for submitting your article "Highly parallelized droplet cultivation and prioritization on antibiotic producers from natural microbial communities" for consideration by *eLife*. Your article has been reviewed by three peer reviewers, and the evaluation has been overseen by Bavesh Kana as the Senior and Reviewing Editor. The following individuals involved in review of your submission have agreed to reveal their identity: Neeraj Dhar (Reviewer #1); Bianca Sclavi (Reviewer #3).

Summary:

In this manuscript, Maler et al. describe a potentially useful microfluidic platform that simplifies isolation and culture of single microbial species from complex environmental samples and to screen them for novel secreted antimicrobial molecules.

Key findings:

1) The authors establish a microfluidic system for culturing bacteria that uses droplets to minimize over-representation of any single population and to facilitate growth of all species present in the samples. The authors indicate that up to 60% of taxa may be missed using standard culture approaches.

2) After the cultivation, the droplets are either deposited on solid medium to enable growth of the microorganisms or are combined with reporter strains to screen for antimicrobial activity. Using this approach, the authors demonstrate a higher and more diverse recovery of bacterial species when compared to standard plating techniques.

3) The authors then screen cultivated the strains for the production of potential antibiotics by injecting reporter strains into the droplets and monitoring their survival. LC-MS analysis of these compounds lead to the identification of previously described antibacterial polyketides, bacillaene A and B and lipophilic molecules.

Conclusion: The platform, whilst not identifying any new chemical matter in the reported screen, provides a novel and tractable approach for cultivating previously unstudied biodiversity and thereby identify new therapeutic agents.

Essential Revisions:

1) The microbial diversity presented is based on the 16S amplicon sequencing of samples after 28 days of incubation. This does not reveal how this system impacts the cultivation of fast-growing vs. slow growing species. The fast-growing species may need only fewer days to grow and reach stationary phase earlier. Whereas slow growing species may have been favoured in this system because they take longer to deplete nutrition in the droplets. When the droplets are used for screening, fast growers may not be metabolically active when compared to slow growers. It is possible that the metabolites produced by fast growers may be sufficient to inhibit the pathogen in question. However, authors also say that the droplets allow diffusion of metabolites (–Discussion). If that is the case, metabolites from fast growing bacteria may have been lost by the time screening for antibiotics is done. More data is needed to clarify this. Could the authors image the droplets over time? For example, weekly data on the growth kinetics? Do the authors have sequencing information of the droplet communities over time? If weekly samples are sequenced, it should give clarity about the impact of the system on species diversity.

2) What fraction of species in the droplet is viable after 28 days incubation. Knowing this is important know whether the inhibition of the pathogen is solely because of direct bacterial competition, by metabolites or combination of both. A simple experiment that may answer this could be the measurement of ATP produced by growing species in the droplets.

3) The strength of the system is that the cultivated droplets could be used for screening of antibiotic producing species. This approach possibly could be used for screening of other metabolite production as well. However, the leakage of metabolites produced by grown species in the metabolites is a drawback in the screening step. Could this issue be solved by sorting the droplets to another bulk vessel? This should remove diffused metabolites and should reveal inhibition by live bacteria vs. accumulated metabolites in the droplets. Can the authors comment on this?

4) The soil samples after culturing in droplets are plated on solid agar medium to allow growth and characterization. Considering that some of the microorganisms that have grown in the droplets are probably incapable of growing on solid media, maybe it could be useful to inoculate the droplets into microwells containing liquid recovery medium for improved recovery and follow-up characterization of the strains. Was this done? If not, please insert some comments/discussion around solid versus liquid culturability.

Reviewer #1:

With the increasing incidence and spread of antibiotic resistant strains, discovery of new antibiotics is urgently needed. A significant number of antibiotics currently in use are natural products that are part of the antibacterial arsenal of microorganisms found in nature. Most of these were discovered from rather readily culturable soil microorganisms. However, recent reports suggest that only a very small fraction of the microbiome diversity found in nature is culturable under standard growth conditions. Against this backdrop, this study provides a methodology that allows better cultivation of this bacterial diversity. The authors use a droplet-microfluidic platform to isolate and cultivate the bacterial cells in droplets. After the cultivation period, the droplets are either deposited on solid medium to enable growth of the microorganisms or are combined with reporter strains to screen for antimicrobial activity. Using this approach the authors demonstrate a higher and more diverse recovery of bacterial species when compared to standard plating techniques. Finally they screen the strains for production of antibiotic compounds by injecting reporter strains into the droplets. One isolate showing potent activity was then subjected to detailed characterization. LC-MS analysis of these compounds lead to their identification as previously described antibacterial polyketides, bacillaene A and B and the lipophilic molecules, surfactins. While these molecules have been reported earlier, it demonstrates the potential of the screening platform to identify new classes of antimicrobial compounds.

The approach and platform is quite exciting so it was a bit disappointing that after all the extensive characterization of the active compounds, it led to the identification of previously known antimicrobial species. Despite lacking any novel scientific findings, the platform itself is useful as it combines and simplifies several different steps required for isolation of environmental microbial species as well as antimicrobial compounds.

Reviewer #2:

Cultivation of microbes is a requirement for functional exploration of any species. However, cultivation of bacteria is a hard problem. This paper describes the use of a microscale droplet based cultivation system that allows cultivation of increased bacterial diversity when compared to plate based system. The microdroplet system allows long incubation times and cultivation of species based on dilution to extinction principle. The dilution to extinction reduces mixed species growth (less than 7% droplets as per the data presented). Cultivated droplet allow diffusion of metabolites that may allow cross talk of metabolites between droplets. This may also help to increase retrieval of bacterial diversity. The cultivation system is conveniently combined with a screening system for antibiotic producing species. While the system overall appears attractive, several points needs to be clarified to better understand the results presented.

Reviewer #3:

Mahler et al. describe an experimental approach using microdroplets to isolate and grow the different bacterial strains that can be found in a complex environment such as a soil sample. The advantage of this approach is that, probably due to the lack of competition and thanks to growth in a liquid environment, it is possible to enrich for strains that are not found when growth is taking place on a solid agar plate. For example, they show that about 60% of the taxa found after growth in the droplets are not found when the same sample is grown on agar. The enrichment of these rare taxa can be used to identify novel antibiotic producer strains. The authors provide an experimental approach using Gram positive and Gram negative reporter strains to screen the droplets for the presence of growth inhibitors within specific droplets isolated via on-chip sorting. Finally, the authors were able to increase the production titer of the natural products to identify possible growth inhibitor molecules by mass spectrometry. In my opinion the conclusions of this paper are supported well by the data. The approach outlined here will be very useful for a wide range of researchers interested in the characterization of complex bacterial communities, not only in soil, but also other environments.

---

## [Author Response]

Essential Revisions:1) The microbial diversity presented is based on the 16S amplicon sequencing of samples after 28 days of incubation. This does not reveal how this system impacts the cultivation of fast-growing vs. slow growing species. The fast-growing species may need only fewer days to grow and reach stationary phase earlier. Whereas slow growing species may have been favoured in this system because they take longer to deplete nutrition in the droplets. When the droplets are used for screening, fast growers may not be metabolically active when compared to slow growers. It is possible that the metabolites produced by fast growers may be sufficient to inhibit the pathogen in question. However, authors also say that the droplets allow diffusion of metabolites (Discussion). If that is the case, metabolites from fast growing bacteria may have been lost by the time screening for antibiotics is done. More data is needed to clarify this. Could the authors image the droplets over time? For example, weekly data on the growth kinetics? Do the authors have sequencing information of the droplet communities over time? If weekly samples are sequenced, it should give clarity about the impact of the system on species diversity.

We are aware that our system might favor slow growing species. We even meant to do that as a mean of dereplication. Fast growing species are usually readily culturable and have been extensively screened in standard natural product screenings. Since the goal of our miniaturized droplet cultivation and screening was to make a new subset of the naturally occurring bacterial diversity available for screens, we specifically decided to choose conditions like low nutrient amounts and long incubation times that might help to culture less known species.

To follow the ratio of fast growers to slow growers, we monitored the growth of bacteria over time in droplets microscopically by determining the relative abundance of droplets with bacterial colonies inside as depicted in Figure 1—figure supplement 2. Figure 1B shows an example for images derived in the counting procedure.

Fast growing species will still be detected by the 16S analysis after 28 days unless they completely lysed, which is generally unlikely. We sequenced after 28 days to get an understanding of the entirety of species diversity that grows in droplets without differentiation between fast- and slow growers. Our goal was to compare for the first time the overall cultivation outcome obtained in droplets and on agar plates. Indeed, for scientific interest, it would have been reasonable to sequence weekly sub-samples to differentiate fast from slow growers, but this was not in our focus.

We agree that some fast growers might not be metabolically active in droplets after 28 d anymore and in consequence might not be successfully isolated and upscaled, even though they might have produced antimicrobial substances. However, as just highlighted, fast growers have very likely been cultivated and screened before. For future work, we even discuss to deselect for fast growers by image-based droplet sorting within the first week to further reduce the chance of rediscovery of natural products. Also, even if the fast growers are not viable anymore, we would still benefit from the information that reporter cells have been inhibited by accumulated metabolites in combination with the genetic information of the producer strains (similar as for the cultures, which we lost again soon after isolation). The genetic information for metabolically inactive cells or species could be obtained by sorting droplets with inhibited reporters and depositing single droplets in wells for single cell genomics, even if they cannot be cultivated outside of droplets (anymore).

We are aware of the mentioned risk that active metabolites might not lead to an inhibition because leakage of the metabolites over droplet borders may dilute the concentration too much and this risk of course increases with time. On the other hand, this risk is higher for the fast growers than for the slow growers (i.e., is not so relevant for the group of interest). However, the droplet border also is not as permeable to as many metabolites as we might have made it appear. To clarify that matter, we included further information in the manuscript (see below). The more hydrophilic a metabolite, the less likely it is partitioning from the aqueous into the oil phase. This means, we might have a bias towards hydrophilic antibiotics in our screening, which could even be a strength, since the outer membrane of Gram-negative bacteria makes them resistant to a number of hydrophobic antimicrobials.

In conclusion, we agree that our screening is biasing for slow growers. We intended to do so, hypothesizing to find more rare and previously not screened species in the fraction of slow growers. Regarding the characterization of the cultivation outcome, we intended to describe the in-droplet cultivation in comparison to standard plate cultivation without distinguishing into slow and fast growers.

To increase the informative value of the manuscript on this topic and to further clarify these points, we included the following lines:

“We also specifically chose conditions like low nutrient amounts and long incubation times that might favor growth and metabolic activity of slow growing species over fast growing species. […] In future work, we even consider to actively deselect for fast growers by image-based droplet sorting within the first week to further reduce the chance of rediscovery of natural products.”

“It is noteworthy that the surfactant-stabilized droplet boundary, though forming a mechanical barrier to prevent cell cross contamination, is not completely abolishing molecular transport for all types of molecules. […] Interdroplet transport of molecules might also present an advantage in the cultivation phase. Allowing communication between species might preserve to some extent the growth-stimulating influence of multispecies communities.”

2) What fraction of species in the droplet is viable after 28 days incubation. Knowing this is important know whether the inhibition of the pathogen is solely because of direct bacterial competition, by metabolites or combination of both. A simple experiment that may answer this could be the measurement of ATP produced by growing species in the droplets.

To avoid inhibition by nutrient competition/depletion, we have added the reporter strains in nutrient rich medium to the droplets, which is stated in the Discussion. We ensured while establishing the reporter assay that reporter cells bring with them enough nutrients to grow and produce a strong signal when a non-producer strain was encapsulated in the droplet before and co-consumed nutrients. Other ways of direct bacterial out competition might indeed be detected by the system, but might be also worth to investigate further. To confirm inhibition, the pool of resulting isolates can be tested with the reporter strain again in 96-well plates.

For our droplets, we are currently not able to read out ATP production and presence of cells at the same time reliably and in high throughput. We would need to have a high throughput method to detect the presence of bacteria independently of metabolic markers, because we would be interested in the ratio of metabolically active cells (ATP producing) to all cells (including metabolically inactive cells). General cell staining methods like membrane dyes are usually not giving intense enough readout signals for the inherent high droplet throughput. A reliable method compatible with our droplet system would be automated image analysis for marker free detection of bacterial colonies. Due to the simultaneous presence of soil particles in droplets, we could not yet develop an image analysis algorithm that distinguishes successfully between cells and soil particles. We are working on such an image analysis for follow up experiments.

3) The strength of the system is that the cultivated droplets could be used for screening of antibiotic producing species. This approach possibly could be used for screening of other metabolite production as well. However, the leakage of metabolites produced by grown species in the metabolites is a drawback in the screening step. Could this issue be solved by sorting the droplets to another bulk vessel? This should remove diffused metabolites and should reveal inhibition by live bacteria vs. accumulated metabolites in the droplets. Can the authors comment on this?

a) Leakage of metabolites is not as extensive as is assumed here. Only fairly hydrophobic or very small molecules will leak into the oil phase. We added information on that to an existing paragraph:

“It is noteworthy that the surfactant-stabilized droplet boundary, though forming a mechanical barrier to prevent cell cross contamination, is not completely abolishing molecular transport for all types of molecules. […] Allowing communication between species might preserve to some extent the growth-stimulating influence of multispecies communities.”

b) If metabolites leak, they leak constantly according to their partitioning coefficient for the water/oil interphase. Even after transferring droplets to another bulk vessel and continuing incubation, the (low) risk of droplet cross contamination would remain.

c) The oil surrounding the droplets works as bridge for leaking metabolites in a cross-over to other droplets. It is continuously pumped through the droplets in the incubation device, resulting in a constant droplet mixing. The comparatively large amount of oil used during incubation (~7 mL oil vs. ~200 pL of a single droplet, or for instance 1.8 µL of droplets when assuming that 1 in 1000 droplets contains a high producer) dilutes any metabolite and makes it unlikely that leaking metabolites have a chance to accumulate in a random other droplet to a minimum inhibitory concentration, especially since droplets change their neighboring droplets constantly. Theoretically, it would also be possible, to regularly exchange the oil phase completely during incubation, however, because of the high intrinsic dilution effect, we don’t think this is necessary. Additionally, our inoculation density to reach single cell inoculations results in ~80 % of droplets being empty, which again lowers the chance of any effective levels of interdroplet transfer. We hypothesize therefore that the risk of missing metabolites due to leakage is much higher than false positively selecting droplets with inhibited reporter due to metabolite accumulation away from the producer. We have summarized these thoughts in the manuscript (see copy above).

4) The soil samples after culturing in droplets are plated on solid agar medium to allow growth and characterization. Considering that some of the microorganisms that have grown in the droplets are probably incapable of growing on solid media, maybe it could be useful to inoculate the droplets into microwells containing liquid recovery medium for improved recovery and follow-up characterization of the strains. Was this done? If not, please insert some comments/discussion around solid versus liquid culturability.

We agree that deposition of droplets in microwells with liquid medium is favorable and we are working on that. However, the technical setup is more complex and needs more optimization to guarantee that only a single droplet is deposited in a well. We extended an already existing paragraph in the Discussion to increase comprehensibility.

"Nevertheless, the overall taxonomic distribution found after in-droplet cultivation could not be recovered with our droplet depositioning and isolation procedure on agar plates. […] Consequently, we are currently developing other upscaling strategies into liquid culture like droplet deposition in microtiter plate-wells"

We also included another idea to achieve strain domestication in a liquid environment:

“Furthermore, we here have isolated strains after one incubation round in droplets but alternatively, the emulsion can be broken under controlled conditions and a new droplet cultivation can be started from the pooled cell suspension. By passaging a community several times through the in-droplet cultivation, replicating organisms become enriched, and the chance of strain domestication increases [Bollmann, Lewis and Epstein, 2007].”